# Efficacy of Pirfenidone and Nintedanib in Interstitial Lung Diseases Other than Idiopathic Pulmonary Fibrosis: A Systematic Review

**DOI:** 10.3390/ijms24097849

**Published:** 2023-04-25

**Authors:** Francesco Amati, Anna Stainer, Veronica Polelli, Marco Mantero, Andrea Gramegna, Francesco Blasi, Stefano Aliberti

**Affiliations:** 1Department of Biomedical Sciences, Humanitas University, Via Rita Levi Montalcini 4, Pieve Emanuele, 20072 Milan, Italy; 2Respiratory Unit, IRCCS Humanitas Research Hospital, Via Manzoni 56, Rozzano, 20089 Milan, Italy; 3Respiratory Unit and Cystic Fibrosis Adult Center, Fondazione IRCCS Ca’ Granda Ospedale Maggiore Policlinico, 20122 Milan, Italy; 4Department of Pathophysiology and Transplantation, Università degli Studi di Milano, 20122 Milan, Italy

**Keywords:** interstitial lung diseases, pirfenidone, nintedanib, forced vital capacity, efficacy

## Abstract

Pirfenidone and nintedanib are antifibrotic medications approved for idiopathic pulmonary fibrosis treatment by regulatory agencies and available for clinical use worldwide. These drugs have been shown to reduce the rate of decline in forced vital capacity and the risk of acute exacerbation among patients with idiopathic pulmonary fibrosis. Recent data suggest that different interstitial lung diseases with a progressive pulmonary fibrosis phenotype can share similar pathogenetic and biological pathways and could be amenable to antifibrotic therapies. Indeed, historical management strategies in interstitial lung disease have failed to identify potential treatments once progression has occurred despite available drugs. In this systematic review, we summarized data on the efficacy of pirfenidone and nintedanib in interstitial lung diseases other than idiopathic pulmonary fibrosis as well as ongoing and upcoming clinical trials. We identify two well-designed trials regarding nintedanib demonstrating the efficacy of this drug in slowing disease progression in patients with interstitial lung diseases other than idiopathic pulmonary fibrosis. On the other hand, results on the use of pirfenidone in interstitial lung diseases other than idiopathic pulmonary fibrosis should be interpreted with more caution on the basis of trial limitations. Several randomized control trials are underway to improve the quality of evidence in the interstitial lung disease field.

## 1. Introduction

Interstitial lung diseases (ILDs) represent a heterogeneous group of more than 200 entities of either known or unknown etiology [1]. Idiopathic pulmonary fibrosis (IPF) is the most common cause of idiopathic ILDs, and it is characterized by progressive fibrosis of the lungs with a poor prognosis [2]. Pirfenidone and nintedanib are antifibrotic medications approved for IPF treatment by regulatory agencies and available for clinical use worldwide [3]. These drugs have been shown to reduce the rate of decline in forced vital capacity (FVC) and the risk of acute exacerbation among patients with IPF [4,5]. Other ILDs may also show a progressive pulmonary fibrosis (PPF) phenotype [6]. Recent data suggest that different ILDs with a PPF phenotype can share similar pathogenetic and biological pathways and could be amenable to the same treatment [7,8]. Thus, it is biologically reasonable that pharmacological agents with antifibrotic properties, such as pirfenidone and nintedanib, may be efficacious in non-IPF PPF and fibrotic ILD [8]. Recent well-designed randomized control trials (RCTs) have confirmed this hypothesis [9,10]. In this review, we summarized data on the efficacy of pirfenidone and nintedanib in ILDs other than IPF as well as ongoing and upcoming clinical trials for these drugs in ILDs.

## 2. Materials and Methods

### 2.1. Search Methodology

Two investigators (AS and FA) independently performed a PubMed search and assessed the studies according to predefined criteria. Reference lists of the selected manuscripts were also manually assessed. The English language restriction was applied. This systematic revision was conducted according to the PRISMA statement [11]. A search of ClinicalTrials.gov was also performed. Conditions used in the search were interstitial lung disease and pulmonary fibrosis; the limits were adults (equal to or more than 18 years old); interventions were nintedanib, pirfenidone, and BIBF 1120. Data from ClinicalTrials.gov were cross-tabulated with the above search. A search of PROSPERO was performed for meta-analyses including interstitial lung diseases as a subject. However, no analyses were identified that covered the same subject as the current systematic review.

### 2.2. Study Selection

We included studies published up to 15 October 2022. Key terms included: (“lung diseases, interstitial” [MeSH Terms]) OR (“lung” [All Fields] AND “diseases” [All Fields] AND “interstitial” [All Fields]) OR “interstitial lung diseases” [All Fields] OR (“interstitial” [All Fields] AND “lung” [All Fields] AND “disease” [All Fields]) OR “interstitial lung disease” [All Fields]) AND “nintedanib” [All Fields]) OR (“pirfenidone”[Supplementary Concept] OR “pirfenidone” [All Fields]).

### 2.3. Data Extraction

After the literature search, titles and abstracts were reviewed by two independent investigators (AS and FA), and in case of disagreement, a final decision was taken by an independent investigator (SA). Articles were included if: (1) the study includes reference to the use of pirfenidone or nintedanib in interstitial lung diseases in adult subjects; (2) the abstract reports the results of a trial. Articles were excluded if: (1) they were written in languages other than English; (2) they were case reports, case series, study designs, comments, or letters to the editors; (3) they were in animal or laboratory models; (4) the study was conducted only in IPF patients; (5) the full text was unavailable; (6) the results were multiple publications from the same study (secondary analysis); (7) they only assessed the pharmacokinetics or safety profile of drugs. Full-text papers fulfilling the above criteria were obtained. Additional studies were found by searching the reference lists of previously published non-randomized studies and systematic reviews.

### 2.4. Data Analysis

Data of interest included the name of the first author, journal and year of publication, study design, number of patients, type of ILD in which the study was performed, inclusion criteria, antifibrotic type, primary outcome, and side effects. Corresponding authors were contacted if data were not present or were unclear in the full text. Given the high degree of heterogeneity across the considered papers, a meta-analysis was not performed.

### 2.5. Critical Assessment of Evidence Quality

Each publication was assessed using the Grades of Recommendation, Assessment, Development, and Evaluation (GRADE) criteria by one of two reviewers (FA), independently checked, and then agreed upon by all authors [12]. GRADE assessments were conducted to assign the quality of the evidence from each reference as high, moderate, low, or very low according to factors that include the study methodology, consistency and precision of the results, and directness of the evidence.

## 3. Results

### 3.1. Summary of the Main Results

In total, 1724 publications were identified in the PubMed searches (Figure 1). Screening of titles and abstracts resulted in 1626 being excluded, leaving 98 publications. A manual search of the references of these selected articles found no additional publications that met the inclusion criteria and were not identified in the PubMed search. The majority of the studies were rejected because they either included IPF patients (n = 71), had multiple publications for the same study (n = 8), had study designs, comments, or case series (n = 5), or were pharmacokinetic or safety profile studies (n = 4).

A final pool of 10 studies was included in the systematic review, with a total of 1990 patients enrolled (Table 1) [9,10,13,14,15,16,17,18,19,20]. Eight studies reported data on patients treated with pirfenidone, while two studies reported data on patients treated with nintedanib. Selected papers were published from 2002 to 2022, with a high frequency in the period 2019–2022 (8/10, 80%) [9,10,13,14,15,16,17,18]. The majority had a randomized, double-blind, placebo-controlled design (8/10, 80%) [9,10,13,15,16,17,19,20]. All papers considered FVC as the primary outcome: eight as the only primary outcome and two in combination with other parameters as a composite outcome. FVC was evaluated at 6 months in three papers, at 1 year in six papers, and at different time endpoints in one paper. A consensus on quality assessment was achieved. Only two studies, both on nintedanib, were evaluated as “high” using the GRADE scale. All other studies were rated as “low” or “very low” using the GRADE scale.

### 3.2. Efficacy of Ninitedanib

Data on nintedanib efficacy in ILD patients other than IPF comes from two large trials, INBUILD and SENSCIS [9,10].

The INBUILD trial enrolled 663 subjects with chronic fibrosing ILDs other than IPF meeting criteria for ILD progression. Patients were randomly assigned (1:1) to receive 150 mg of oral nintedanib twice daily or a placebo for at least 52 weeks. [9]. The term “progressive fibrosing ILDs” (PF-ILDs) refers to a spectrum of lung disorders other than IPF that share a progressive clinical phenotype that is characterized by an increasing extent of fibrosis on high-resolution CT, a decline in lung function, and worsening of symptoms despite management deemed appropriate in clinical practice [6]. The result of the trial showed that the FVC decline was −187.8 mL per year with placebo and −80.8 mL per year with nintedanib (*p* < 0.001), resulting in a difference of 107 mL. The reduction in annual FVC decline was similar to the rate observed in IPF nintedanib trials (125.3 mL in INPULSIS-1 and 93.7 mL in INPULSIS-2), suggesting a similar biological effect [4,9]. In the INBUILD trial, glucocorticoids were taken by over half the patients at baseline, while 15.2% were taking immunomodulatory therapies [9]. The effect of nintedanib on reducing FVC decline was not influenced by the use of glucocorticoids and immunomodulatory therapies [21]. Several different types of ILD (other than IPF) were included in INBUILD and classified into five subgroups: hypersensitivity pneumonitis, idiopathic non-specific interstitial pneumonia, unclassifiable ILD, autoimmune disease–related ILD, and “other” fibrosing ILDs. In a post-hoc analysis, no significant differences in efficacy between disease subgroups were observed [22].

The SENSCIS trial was a randomized, double-blind, placebo-controlled trial in which 576 patients with systemic sclerosis (SSc)-ILD were randomly assigned (1:1) to receive 150 mg of oral nintedanib twice daily or a placebo for at least 52 weeks [10]. SENSCIS enrolled adults fulfilling the American College of Rheumatology (ACR)/European League Against Rheumatism (EULAR) classification criteria for SSc with the onset of the first non-Raynaud’s symptom within the 7 years prior to screening. SSc-ILD was confirmed by a high-resolution computed tomography (HRCT) scan performed within the 12 months prior to screening, which showed fibrotic ILD to affect ≥10% of the lungs, as assessed by central review. The observed FVC decline was 52.4 mL per year in the nintedanib group and −93.3 mL per year in the placebo group (*p* = 0.04).

### 3.3. Efficacy of Pirfenidone

Although more trials on pirfenidone’s efficacy in ILD other than IPF patients have been published compared to nintedanib, the quality of the evidence is lower [13,14,15,16,17,18,19,20].

The largest trial that has been published so far on pirfenidone efficacy is the UILD study [16]. In this randomized, double-blind, placebo-controlled, 1:1, phase 2 trial, 253 patients with a progressive, fibrosing, unclassifiable ILD (uILD) were randomized to pirfenidone 2403 mg daily or placebo. The mean predicted change in FVC from baseline over 24 weeks was −87.7 mL in the pirfenidone group versus −157.1 mL in the placebo group (*p* = 0.002). Notably, the subgroup analysis suggests that pirfenidone may be less effective in uILD patients receiving mycophenolate at randomization, whereas a beneficial treatment effect for pirfenidone on FVC change was observed in patients not receiving MMF at randomization, regardless of previous corticosteroid use [23]. Since uILD represents a diagnosis of exclusion, a post-hoc analysis of data from the pirfenidone in the UILD trial was performed based on the surgical lung biopsy (SLB) status [24]. The study revealed that pirfenidone may be an effective option regardless of SLB status.

The RELIEF study was conducted in ILD patients with progressive functional decline despite conventional therapy [15]. In this randomized, double-blind, placebo-controlled, parallel Phase 2b trial, 127 patients were randomized to receive pirfenidone 2403 mg daily or a placebo. The absolute change in percentage of FVC% predicted from baseline to week 48 in the intention-to-treat population was evaluated as the primary outcome. Pirfenidone-treated patients had a significantly lower decline in FVC% predicted compared with placebo-treated ones (*p* = 0.043). However, due to the premature trial termination due to slow recruitment and the subsequent issue related to underpowering, interpreting FVC trends in the RELIEF study should be done with caution.

TRAIL1 was a randomized, double-blind, placebo-controlled, multicenter phase 2 trial conducted in patients with RA-ILD [13]. Patients were randomly assigned (1:1) to receive 2403 mg oral pirfenidone (pirfenidone group) or placebo (placebo group) daily. The primary endpoint was the incidence of the composite endpoint of a decline from baseline in predicted forced vital capacity (FVC%) of 10% or more or death during a 52-week treatment period. The difference in the proportion of patients who met the composite primary endpoint between the two groups was not significant (11% patients in the pirfenidone group vs. 15% in the placebo group; OR 0.67 [95% CI 0.22 to 2.03]; *p* = 0.48). However, the trial was stopped early due to slow recruitment and the COVID-19 pandemic.

A double-blind, randomized, placebo-controlled pilot study has been conducted in SSc-ILD by Acharya et al. [17]. Patients were randomized to receive either pirfenidone or a placebo for 6 months. The primary outcome was the proportion of subjects with either stabilization or improvement in FVC at 6 months. In this study, stabilization/improvement in FVC was seen in 16 (94.1%) and 13 (76.5%) subjects in the pirfenidone and placebo groups, respectively (*p* = 0.33). However, the small sample size and the short follow-up period limit the interpretability of the results.

Recently, a prospective, controlled cohort, single-center study was conducted in patients with CTD-ILD [14]. Physicians recommended whether to add pirfenidone to background therapy (glucocorticoids and/or immunosuppressive therapy) and solicited the opinions of patients according to the inclusion criteria. The primary endpoint of the study was the change in FVC and DLCO after 24 weeks of treatment according to the 4 CTD-ILD groups. The authors found a significant improvement in FVC% in the pirfenidone group in the case of SSc-ILD, or inflammatory myopathy-ILD. The DLCO% was significantly improved in RA-ILD patients treated with pirfenidone compared to the control case. However, the study suffers from several methodology limitations (single-center study; limited sample size of each distinct CTD-ILD group; lack of randomized control arms; short duration of follow-up) that underpower the results.

An open-label, proof-of-concept, randomized, single-center study was conducted in CHP patients to evaluate the efficacy of pirfenidone added to immunosuppressive drugs [18]. The primary outcome was the change in predicted FVC value in % and ml after 12 months. The results showed that the inclusion of pirfenidone was not associated with a significant improvement in the predicted value of FVC (% and ml). However, results should be interpreted with caution due to the small sample size.

Finally, two small, randomized, double-blind, placebo-controlled trials have been conducted in Hermansky-Pudlak syndrome, a rare, genetic, multisystem disorder characterized by oculocutaneous albinism, bleeding diathesis, immunodeficiency, granulomatous colitis, and pulmonary fibrosis with a similar presentation to IPF [19,20,25]. Both trials assessed the rate of change in FVC between the placebo and pirfenidone groups as the primary outcome. The timing of the FVC assessment varied depending on the trial. Notably, in only one of the two trials, a statistically significant difference was observed [20].

## 4. Discussion

The use of antifibrotic agents in ILD has been a topic of worldwide interest in the last few years. In this systematic review, we gathered data regarding the efficacy of pirfenidone and nintedanib in ILD other than IPF. Data on nintedanib efficacy in ILD patients other than IPF comes from two high-quality trials, INBUILD and SENSCIS [9,10]. Both trials have shown that nintedanib is efficacious in attenuating disease progression in patients with non-IPF ILD, despite management and regardless of the radiographic pattern of fibrosis.

However, the five major diagnostic subgroups identified in the INBUILD trial are underpowered, and FVC treatment effects lie immediately above “statistical significance.” Thus, results on specific subgroups in the INBUILD trial should be interpreted with caution. In the SENSCIS trial, treatment with nintedanib slowed down the annual loss of FVC by 40 mL/year compared to placebo, apparently unimpressive compared to the INBUILD trial [9,10]. However, the rate of decline in the placebo arm was also lower compared to the INBUILD trial (−93.3 mL versus 187.8 mL). This mirrors the heterogeneous disease behavior of SSc-ILD, which has a variable course and only becomes progressive in some patients and is characterized by both an increase in fibrotic abnormalities on high-resolution CT and a decline in FVC. Notably, 43% of progression was prevented compared with FVC decline in the placebo arm, leading to rapid regulatory approval for nintedanib in SSc-ILD. Notably, in both SENESCIS and INBUILD trials, curves for FVC change from baseline separated by week 12 and continued to diverge since the end of the study, similarly to nintedanib trials in IPF [4,9,10]. Moreover, the benefit of nintedanib on FVC decline was observed regardless of fibrotic pattern or lung fibrosis extent on HRCT. These considerations are consistent with the overlapping pathophysiology of progressive fibrotic ILDs, irrespective of ILD diagnosis. Conclusions on pirfenidone’s efficacy in ILD other than IPF are more guarded, on the basis of trial limitations. The largest study on pirfenidone in ILD other than IPF, the UILD study, suffers from several limitations. The primary endpoint was assessed with serial home spirometry, which provided less meaningful data compared to the previous published trials on pirfenidone in IPF. Subgroup analysis of the UILD study suggests that pirfenidone may be less effective in UILD patients receiving mycophenolate at randomization, whereas a beneficial treatment effect for pirfenidone on FVC change was observed in patients not receiving MMF at randomization, regardless of previous corticosteroid use [22]. However, the subgroup analysis has been limited by the small sample size. Moreover, uILD represents a diagnosis of exclusion; indeed, patients generally receive a uILD diagnosis when all other ILDs have been ruled out. Centers may apply varying intensities of diagnostic investigations or different diagnostic thresholds when evaluating a patient, leading to heterogeneity in the uILD definition [23]. This is a relevant consideration since uILD patients were identified by investigators at each site and not by a central reading in the UILD trial. Moreover, other limitations in pirfenidone studies should be acknowledged. Interpreting FVC trends in the RELIEF study should be done with caution due to the premature trial termination due to slow recruitment and the consequent issue related to underpowering [15]. Notably, some of the trials concerning pirfenidone were stopped early due to slow recruitment and the COVID-19 pandemic [15,17,18]. The small sample size and the short follow-up period undercut the results of the majority of pirfenidone trials [13,14,17,18,19,20].

Other respiratory and extra-respiratory parameters have been evaluated as secondary outcomes in the studies analyzed. Concerning lung function, there was a significantly lower decline in diffusing lung capacity for carbon monoxide (DLCO) in the group receiving pirfenidone compared to the placebo group, suggesting a beneficial treatment effect of pirfenidone [15,18,19]. However, this was not a consistent finding [16]. Of note, symptoms and quality of life measured by the Saint George Respiratory Questionnaire (SGRQ) were not significantly improved by antifibrotic treatment [9,10,17,19,20]. Likewise, nintedanib did not improve fibrosis-related skin involvement as measured by the modified Rodnan skin score [10].

## 5. Ongoing Clinical Trials

Several interventional clinical trials are underway to explore the use of antifibrotic therapies in both fibrosing ILDs and PF-ILD.

### 5.1. Ongoing Clinical Trials on Nintedanib

Several interventional trials are evaluating the use of the drug and its efficacy in fibrotic ILDs, while no studies are evaluating its use in non-fibrotic ILDs to prevent subsequent fibrosis, except for one trial on COVID-19 sequelae (Table 2) [26,27,28,29,30,31,32,33,34,35,36,37].

Two clinical trials are evaluating the efficacy of nintedanib in patients with pneumoconiosis (NCT05067517, NCT04161014) [27,33].

The NCT05335278 is evaluating the use of nintedanib in patients with myositis (dermatomyositis, polymyositis, overlap myositis, or anti-synthetase syndrome)-related ILD [28]. It is important to note that only patients with PPF-ILD can be included in the trial.

Bronchiolitis obliterans syndrome (BOS) is a severe complication after transplant, and nintedanib has been proposed as a possible therapeutic agent due to its antifibrotic properties. The NINBOST2018 trial (NCT03805477) is enrolling patients with BOS after allogeneic hematopoietic cell transplantation to evaluate safety and tolerability, while the INFINITx-BOS trial (NCT03283007) is a phase III trial assessing the efficacy of nintedanib in reducing the rate of decline of FEV1 in patients with BOS after lung transplant [34,35].

Although not properly a fibrotic ILD, the efficacy of nintedanib is under study in patients with lymphangioleiomyomatosis (LAM) (NCT03062943) [36]. Because of its anti-tyrosine kinase receptor inhibition on platelet-derived growth factor receptor (PDGFR), nintedanib has been shown to inhibit mTOR activation. Moreover, the inhibition of vascular-endothelial growth factor (VEGF), platelet-derived growth factor (PDGF), and fibroblast growth factor (FGF) signaling pathways reduces tumor angiogenesis in the lung and consequently could potentially contribute to the reduction of LAM cell dissemination and the progression of the disease [38,39].

Finally, several studies have suggested that COVID-19 patients can develop pulmonary fibrosis [40]. Four clinical trials (NCT04856111, NCT04619680, NCT04541680, and NCT04338802) are evaluating the efficacy of nintedanib in COVID-19 through changes in FVC as the primary outcome [29,30,31,32]. The inclusion criteria and time from SARS-CoV-2 infection differ between studies. The NINTECOR trial (NCT04541680) is the largest ongoing trial, including 250 patients with a previous hospitalization for COVID-19 with fibrotic lung sequelae [31]. Notably, the ENDCOV-I (NCT04619680) is the only trial including patients with both fibrotic and non-fibrotic ILD after SARS-CoV-2 infection [30].

### 5.2. Ongoing Clinical Trials on Pirfenidone

Unlike nintedanib, pirfenidone is currently approved only for the treatment of IPF. However, several interventional clinical trials (Table 3) are evaluating the efficacy of pirfenidone in different subtypes of ILDs [41,42,43,44,45,46,47,48,49,50,51,52,53,54,55,56,57,58,59].

Several clinical trials are targeting patients affected by pneumoconiosis, particularly asbestosis, coal worker disease, and silicosis (NCT05288179, NCT0513345, NCT05118256, and NCT04461587) [42,43,44,49].

Three clinical trials are targeting patients with HP associated with a PPF phenotype (NCT02958917, NCT04675619), while one targets recently diagnosed CHP patients (NCT02496182) [47,57,58].

One clinical trial (NCT04193592) is assessing the efficacy of pirfenidone in pulmonary fibrosis related to Hermansky-Pudlak syndrome [50].

The STOP-BOS trial (NCT03315741) is evaluating the safety and tolerability of pirfenidone in patients with BOS after hematopoietic cell transplant, while the EPOS study (NCT02262299) is assessing its efficacy in lung transplant recipients with BOS [55,59].

Different clinical trials are evaluating the use of pirfenidone in CTD-ILD (NCT05505409, NCT04928586) and in specific CTDs including dermatomyositis, systemic sclerosis, and ANCA-related vasculitis (NCT03857854, NCT03856853, NCT03221257, and NCT03385668) [41,46,52,53,54,56].

Regarding CTD-ILD, the NCT05505409 trial is recruiting patients with CTDs and ILD unresponsive to treatment, while the NCT04928586 trial is recruiting patients with CTD-ILD to evaluate the efficacy of pirfenidone associated with disease-modifying antirheumatic drugs (DMARDs) against DMARDs alone [41,46]. Interestingly, both trials are including not only patients with definite CTDs but also those with undifferentiated connective tissue disease (UCTD)/ interstitial pneumonia with autoimmune features (IPAF). Note that both studies do not require progressive pulmonary fibrosis as an inclusion criterion.

Regarding specific CTDs, the NCT03857854 trial has enrolled patients affected by dermatomyositis, including patients with fibrotic and non-fibrotic ILD [52]. The SLS III trial (NCT03221257) is the only trial evaluating the efficacy of mycophenolate in association with pirfenidone against a placebo [56].

The PIONEER trial (NCT05075161) is evaluating the use of pirfenidone to prevent pulmonary fibrosis in patients admitted to an intensive care unit due to acute respiratory distress syndrome (ARDS) [45]. Only patients with moderate or severe ARDS and an inflammatory phenotype can be included in the study.

One trial (NCT03902509) is evaluating the efficacy of pirfenidone in the treatment of grade 2–3 pulmonary radiation injury [51].

Finally, only one clinical trial is evaluating pirfenidone in patients with at least 5% post-COVID-19 pulmonary fibrosis (NCT04607928) [48].

## 6. Conclusions

This systematic review focused on consolidating findings from the literature on the effects of pirfenidone and nintedanib on patients with non-IPF ILDs. Regarding pirfenidone, the quality of the evidence ranges from very low to low. Results of the included studies suggest that pirfenidone may have a beneficial impact on lung function in patients with non-IPF ILD. In particular, the unclassifiable ILD patients might benefit from pirfenidone treatment. Regarding nintedanib, the quality of the evidence is considered high according to the GRADE criteria, although only 2 studies have assessed efficacy in non-IPF ILD. The overall findings suggest that nintedanib may have a beneficial impact on disease progression in patients with non-IPF ILD. However, results on both drugs should be interpreted with caution because of limitations in the available evidence. Moreover, there are several controversial points that should be clarified with further studies and evidence. Examples of challenges that need to be addressed in the future are the timing of therapy initiation and the strategies that should be adopted for overlap or combination with existing immunosuppressive therapies and potential drug interactions. Several RCTs are underway to improve the quality of evidence in the ILD field.

## Figures and Tables

**Figure 1 ijms-24-07849-f001:**
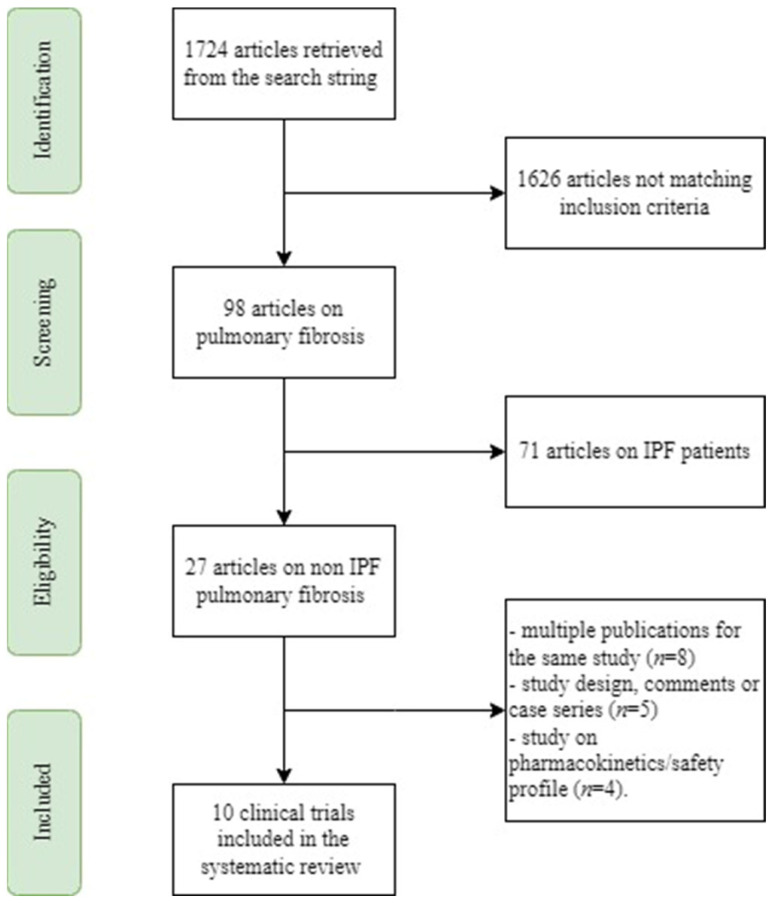
Flow chart of the systematic review.

**Table 1 ijms-24-07849-t001:** Antifibrotic trials in ILDs beyond IPF [9,10,13,14,15,16,17,18,19,20].

Article	Study Design	Sample Size	Type of ILD	Inclusion Criteria	Intervention	Primary Outcome	Results	Side Effects	GRADE
Solomon JJ; Lancet Resp Med 2022 [13]	Randomized, double-blind, placebo-controlled, 1:1, phase 2 trial	123	RA-ILD	ILD diagnosis on a HRCT scan and, when available, a lung biopsy.	Pirfenidone, 2403 mg oral daily, or placebo	Composite endpoint of a decline from baseline FVC% of 10% or more or death during the 52-week treatment period	11% of patients in the pirfenidone group vs. 15% of patients in the placebo group; (*p* = 0.48).	There was no significant difference in the rate of serious AEs between the two groups.	Low
Wang J; Front Med (Lausanne) 2022 [14]	Non-randomized, single-center, 1:1, controlled cohort study	136	CTD-ILD	FVC < 80%; DLCO < 80% No pulmonary fibrosis improvement with glucocorticoid and/or immunosuppressant treatment.	Pirfenidone, 1800 mg/day, or placebo	Change in FVC and DLCO after 24 weeks of treatment according to 4 groups (SSc; RA; inflammatory myopathy; other CTD)	FVC% in the SSc-pirfenidone group was improved by 6.60%, while this value was 0.55% in the SSc-non-pirfenidone group (*p* = 0.042). The elevation in FVC% was also different between the pirfenidone and control groups of patients with inflammatory myopathy: 7.50% vs. 1.00% (*p* = 0.016). The DLCO% of RA-pirfenidone was enhanced by 7.40% compared with RA-non-pirfenidone which decreased by 5.50% from baseline (*p* = 0.002).	No differences were found in terms of total AEs between the two groups. Gastrointestinal events were more often found in the pirfenidone group than the control group.	Very Low
Behr; Lancet Respir Med 2021 [15]	Randomized double-blind, placebo-controlled, 1:1, 2b trial	127	CTD-ILD, NSIP, CHP, and asbestosis-ILD	FVC 40–90%; DLCO 10–90%; Annual decline of FVC of at least 5% despite conventional therapy.	Pirfenidone, 2403 mg daily, or placebo	Absolute change in percentage of FVC % predicted from baseline to week 48 in the intention-to-treat population	There was significantly lower decline in FVC% predicted in the pirfenidone group compared with placebo group (*p* = 0.043).	Severe AEs (grades 3–4) of nausea (two patients on pirfenidone, two on placebo), dyspnea (one patient on pirfenidone, one on placebo), and diarrhea (one patient on pirfenidone) were occasionally observed.	Low
Maher; Lancet Resp Med 2020 [16]	Randomized double-blind, placebo-controlled, 1:1, phase 2	253	Progressive fibrosing uILD *	FVC > 45%; DLCO > 30%; Fibrosis affecting at least 10% of the lung volume on HRCT.	Pirfenidone, 2403 mg daily, or placebo	The mean predicted change in FVC from baseline over 24 weeks	−87.7 mL in the pirfenidone group versus−157.1 mL in the placebo group (*p* = 0.002).	The most common TEAEs were gastrointestinal disorders (47% in the pirfenidone group vs. 26% in the placebo group), fatigue (13% vs. 10%), and rash (10% vs. 7%).	Low
Acharya; Rheumatol Int. 2020 [17]	Randomized double-blind, placebo-controlled, 1:1, phases 2–3	34	SSc-ILD	FVC 50–80%.	Pirfenidone, 2403 mg daily, or placebo	Proportion of subjects with either stabilization or improvement in FVC at 6 months	Stabilization/improvement in FVC was seen in 94.1% and 76.5% of subjects in the pirfenidone and placebo groups, respectively (*p* = 0.33).	Common AEs were gastrointestinal disturbances and skin rashes.	Very low
Mateos-Toledo; Arch Bronconeumol 2020 [18]	Open-label, proof-of-concept study	22	CHP	More than 12 months of symptoms before diagnosis; The presence of fibrosis and architectural distortion in the histopathological evaluation of the lung biopsy affected more than 10% of the *parenchima*.	Pirfenidone 1800 mg/day plus prednisone (to reach a maintenance dose of 0.125 mg/kg) plus azathioprine (1–2 mg/kg per day) or prednisone plus azathioprine for 1 year.	Change of predicted value of the FVC in % and ml at 12 months	No significant changes were observed in the predicted value of FVC (% and mL) from baseline to 12 months in any of the groups.	Nausea (27% vs. 0%), diarrhea (27% vs. 11%), and dyspepsia (100% vs. 67%) were more frequent in the pirfenidone group.	Very low
Flaherty; NEJM 2019 [9]	Randomized double blind, placebo controlled, 1:1, phase 3	663	CHP, CTD-ILD, NSIP, uILD, and others	Fibrosing ILDs affecting at least 10% of lung volume on HRCT; ILD Progression **; FVC > 45%; DLCO 30–80%.	Nintedanib, 150 mg twice daily, or placebo	Annual rate of decline in FVC	−80.8 mL per year with nintedanib versus−187.8 per year with placebo (*p* < 0.001)	Diarrhea 66.9% in nintedanib group and 23.9% in placebo group. Abnormalities on liver-function testing were more common in the nintedanib group.	High
Distler; NEJM 2019 [10]	Randomized double blind, placebo controlled, 1:1, phase 3	576	SSc-ILD	Onset of the first non-Raynaud’s symptom within the past 7 years; Fibrosis affecting at least 10% of the lung volume on HRCT.	Nintedanib 150 mg twice daily or placebo	Annual rate of decline in FVC	−52.4 mL per year in the nintedanib group versus −93.3 mL per year in the placebo group (*p* = 0.04)	Diarrhea was reported in 75.7% of the patients in the nintedanib group and in 31.6% of those in the placebo group. AEs led to the permanent discontinuation of the trial drug in 16.0% of the nintedanib group and 8.7% in the placebo group.	High
O’Brien; Mol Genet Metab 2011 [19]	Randomized, double-blind, placebo-controlled trial, 2:1	35	Hermansky-Pudlak syndrome ILD	FVC 51–85%.	Pirfenidone, 1602 mg daily, or placebo	Rate of change in post-bronchodilator FVC at 12 months	There was no statistical difference between the placebo and pirfenidone groups.	Nausea (17% vs. 8%) and photosensitivity rash (9% vs. 0%) were more frequent in the pirfenidone group.	Very low
Gahl; Mol Genet Metab. 2002 [20]	Randomized, placebo-controlled, and 1:1 trial	21	Hermansky-Pudlak syndrome ILD	FVC 40–75%.	Pirfenidone, 800 mg t.i.d., or placebo	Rate of change in FVC at 21, 32, 36, and 44 months	11 pirfenidone-treated patients lost FVC at a rate of 5% of predicted (∼400 mL) per year, slower than 10 placebo-treated patients (*p* = 0.001).	There was no statistical difference between the placebo and pirfenidone groups in terms of AEs.	Very low

Abbreviations: RA: rheumatoid arthritis; ILD: interstitial lung disease; HRCT: high-resolution computerized tomography; FVC: forced vital capacity; AE: adverse events; SSc: systemic sclerosis; CTD: connective tissue disease; DLCO: diffusing capacity for carbon monoxide; NSIP: non-specific interstitial pneumonitis; CHP: chronic hypersensitivity pneumonitis; TEAE: treatment-emergent adverse events; uILD: unclassifiable interstitial lung disease. Notes: * Definition of progressive fibrosing unclassifiable ILD if: Fibrosing ILD that could not be classified with moderate or high confidence into any category of ILD after multidisciplinary team discussion at each center; AND Progressive disease, defined as ≥5% absolute decline in percent predicted FVC or significant symptomatic worsening not due to cardiac, pulmonary (except worsening of underlying unclassifiable ILD), vascular, or other causes (as determined by the investigator) within the previous 6 months. ** Definition of progressive fibrosing lung diseases if at least one of the following in the last 24 months despite treatment: FVC decline > 10%; FVC decline ≥ 5% and decline in DLCO of ≥15%; FVC decline ≥ 5% and increased fibrosis on HRCT; FVC decline ≥ 5% and progressive symptoms; Progressive symptoms and increased fibrosis on HRCT.

**Table 2 ijms-24-07849-t002:** Ongoing clinical trial on nintendanib [26,27,28,29,30,31,32,33,34,35,36,37].

NCT Number	Disease	Phase	Enrollment	Study Design	Inclusion Criteria	Primary Outcome	Secondary Outcomes
NCT05065190 [26]	PF-ILD	3	90	Interventional Randomized Quadruple blind	Progressive fibrosis * Fibrosing lung disease on HRCT FVC ≥ 45%	Change in FVC [Time frame 52 weeks]	N/A
NCT05067517 [27]	Progressive Fibrosing Coal Mine Dust-Induced ILD	3	160	Interventional Randomized Triple blind	30% ≤ DLCO < 80%. FVC ≥ 45%	Change in FVC [Time frame 12–24–36–52 weeks]	Change in pulmonary function Absolute change from baseline in the L-PF Symptoms (cough and dyspnea) domain score Absolute change from baseline in the K- BILD total score Progression on HRCT 6MWT Time to all-cause and respiratory mortality Time for progression
NCT05335278 [28]	Myositis Associated ILD	N/A	25	Interventional Open label	Extent of ILD disease ≥ 10% on HRCT done within 12 months of enrolment Progressive disease within 24 months of the screening visit Current and ongoing treatment with immunosuppressive medications, on a stable medication regimen and dosage for at least 6 weeks (considered standard of care medical therapy)	Tolerability AE [Time frame 24 weeks]	Change in FVC Change in DLCO Change in 6MWD
NCT04856111 [29]	COVID-19	4	48	Interventional Randomized Single blind	Post-COVID parenchymal involvement >10% of the lung parenchyma or having persistent reticulation or persistent consolidation despite a trial of glucocorticoids (minimum prednisolone dose of 10 mg/day, or equivalent) for a minimum period of 4 weeks after discharge for the acute COVID-19 illness	Change in the FVC [Time frame 24 weeks]	Proportion of subjects with FVC improvement or stabilization Change in dyspnoea Change in resting oxygen saturation Proportion of subjects with oxygen desaturation on exercise testing Change in the 6MWD Change the SF-36 and K-BILD questionnaires Changes in HRCT scores AE
NCT04619680 [30]	COVID-19	4	170	Interventional Randomized Triple blind	Required one of the following after diagnosis with SARS-CoV-2: - supplemental oxygen through nasal cannula; - high flow oxygen; - non-invasive ventilation or mechanical ventilation or a history of desaturation below 90%; FVC <91% predicted or DLCO <71%	Change in FVC [Time frame 180 days]	Chest CT visual score Change in the SGRQ, K-BILD, LCQ, and SF-36 questionnaires Change in 6MWT Functional Assessment of Chronic Illness Number of deaths due to any or respiratory cause AE
NCT04541680 [31]	COVID-19	3	250	Interventional Randomized Triple blind	1. History of hospitalization for COVID-19 infection documented with positive PCR or positive serology in the previous 2 to 12 months 2. Lung opacities on HRCT involving > 10% of the lung volume with fibrotic features 3. DLCO ≤ 70%	Change in FVC [Time frame 12 months]	Change in DLCO Change in 6MWT HRCT lung opacities extension Change in health-related quality of life Evolution of dyspnoea over time AE
NCT04338802 [32]	COVID-19	2	96	Interventional Open Label	18–70 years old. CT examination of patients with multiple fibrotic shadows in both lungs.	Change in FVC [Time frame 8 weeks]	Changes in DLCO Changes in the 6MWT Changes in the HRCT score
NCT04161014 [33]	Pneumoconiosis	2	100	Interventional Open Label	1. Pneumoconiosis diagnosis confirmed at the Occupational MDT 2. Diffuse fibrosing lung disease >10% on HRCT with protocol criteria for progression 3. Asbestosis, silicosis, coal worker’s pneumoconiosis, and diffuse dust fibrosis 3. FVC ≥ 45% and DLCO > 30%	Change in FVC [Time frame 36 months]	K-BILD score Time to acute exacerbation Time to referral for lung transplantation Time to death
NCT03805477 [34]	BOS	2	20	Interventional Open Label	Time interval from transplant ≤ 5 years at the time of inclusion Absolute decline of FEV1 ≥ 10% within the past 12 months	AE rate leading to interruption/ discontinuation of study treatment [Time frame 12 months]	Changes in pulmonary function parameters Change in eNO Nitrogen-washout Changes in in 6MWD Cumulative steroid doses Occurrence of GvHD in other organs Disease-free survival of underlying hematologic disease Overall survival
NCT03283007 [35]	BOS	3	80	Interventional Randomized Quadruple Blind	At least 6 months post-lung transplant Progressive BOS **	Change in FEV1 [Time frame 1, 2, 3, 6, 9, 12, and after 13 months]	Exercise tolerance Quality of life improvement Efficacy to hamper FEV1 decrease Efficacy to hamper the progression of BOS Change in oxygen saturation nintedanib tolerance Explanatory parameters of fibrotic pathways
NCT03062943 [36]	LAM	2	30	Interventional Open Label	LAM patients with proven side effects and/or toxicities/contraindications to sirolimus therapy will be eligible for this study.	Change in FEV1 [Time frame 12 months]	AE
NCT02496585 [37]	Radiation induced lung injury	2	33	Interventional Randomized Double Blind	Prior treatment with thoracic radiotherapy completed >4 weeks and ≤9 months prior to enrolment	Number of patients who are free from pulmonary exacerbations [Time frame 12 months]	N/A

Abbreviations: ILD: interstitial lung disease; HRCT: high-resolution computed tomography; CT: computed tomography; FVC: forced vital capacity; DLCO: diffusion capacity for carbon monoxide; 6MWD: six-minute walking distance FEV1: forced expiratory volume in 1 sec. L-PF: living with pulmonary fibrosis K-BILD: King’s Brief Interstitial Lung Disease Questionnaire; 6MWT: six-minute walking test; CTD: connective tissue disease; RT-PCR: real-time polymerase chain reaction mMRC: modified Medical Research Council; ATS/ERS: American Thoracic Society/European Respiratory Society; SGRQ: Saint George Respiratory Questionnaire; LCQ: Leicester Cough Questionnaire; PF-ILD: progressive fibrosing interstitial lung disease; BOS: bronchiolitis obliterans syndrome PFT: pulmonary function test cGVHD: chronic graft-versus-host disease; AE: adverse event; TLC: total lung capacity; SSC-ILD: systemic sclerosis-related interstitial lung disease; LTx: lung transplant ISHLT: International Society for Heart and Lung Transplantation; VC: vital capacity LAM: Lymphangioleiomyomatosis; TSC: Tuberous Sclerosis Complex; VEGFD: Vascular Endothelial Growth Factor D; MDT: multi-disciplinary team; eNO: exhaled nitric oxide. Notes: * Progressive Phenotype within 24 months of screening visit: decline in FVC % ≥10%; decline in FVC % of ≥5–<10% combined with worsening of respiratory symptoms; decline in FVC % of ≥5–<10% combined with increasing extent of fibrotic changes on chest imaging; worsening of respiratory symptoms as well as increasing extent of fibrotic changes on chest imaging. ** a total decline of at least 200 mL in FEV1 in these last 12 months AND FEV1/VC < 0.7. Azithromycin therapy for at least 4 weeks prior to the first visit.

**Table 3 ijms-24-07849-t003:** Ongoing clinical trial on pirfenidone.

NCT Number	Disease	Phase	Enrollment	Study Design	Inclusion Criteria	Primary Outcome	Secondary Outcomes
NCT05505409 [41]	CTD-ILD	4	120	Interventional Open Label	Patients with clinical deterioration more than 1 month after diagnosis of ILD history, or poor response or intolerance to glucocorticoids or immunosuppressants treatment, or poor response or intolerance to other antifibrotic drugs.	Change in FVC% [Time Frame: 6 months]	Changes in pulmonary function parameters Progression-free survival Change in 6MWTD Radiological changes BORG dyspnea index score Changes in inflammatory biomarkers Changes in primary disease activity AE
NCT05288179 [42]	Pneumoconiosis	3	272	Interventional Randomized Quadruple Blind	40% ≥ FVC > 80% 30% ≥ DLCO > 80%	Change in FVC% [Time Frame: 52 weeks]	Change in FVC (L) Change in DLCO%
NCT05133453 [43]	Asbestosis	N/A	40	Interventional Randomized Open Label	FVC ≥ 50% DLCO ≥ 30% Duration since diagnosis at least one year before the study.	FVC % DLCO Radiological findings change [Time frame: 6–12 months]	N/A
NCT05118256 [44]	Silicosis	2	18	Interventional Randomized Single Blind	Age range: 18–65 years. Progressive massive fibrosis due to silicosis.	Metabolic pulmonary activity assessed by PET-CT scan (18 FFDG) [Time Frame: baseline (day 1), 6–12 months]	Cell biomarkers in peripheral blood AE EQ-5D-5L test Respiratory function parameters
NCT05075161 [45]	Post ARDS fibrosis	3	130	Interventional Randomized Quadruple Blind	The inflammatory ARDS phenotype is defined by at least one of the following: - High plasma levels of inflammatory biomarkers; - Vasopressor dependence; - Lower serum bicarbonate or increased serum lactate.	The number of ventilator free days (VFD) [Time frame: 28 days]	ICU-free days at day 28 Cumulative SOFA-free point at day 28 Hospital length of stay Fibroproliferative changes on HRCT Mortality at ICU/hospital discharge Quality of life assessment at follow-up (6–12 months) with SF-36 and EQ-5D scores Percentage change in pulmonary function parameters Proportion of subjects who develop heart dysfunction AE Use of rescue therapies for severe hypoxemia
NCT04928586 [46]	CTD-ILD	4	200	Interventional Randomized Open Label	DMARDs treatment.	Change in FVC Change in DLCO [Time frame: 12 months]	Changes in dyspnea score Imaging changes Changes in 6MWD Changes in CRP and ESR Changes in VAS score AE
NCT04675619 [47]	PF-CHP	2	40	Interventional Randomized Open Label	Progressive fibrosis *	Change in FVC Change in 6MWD [Time frame: 12 months]	N/A
NCT04607928 [48]	COVID-19	2	148	Interventional Randomized Triple Blind	Fibrotic radiological changes ≥ 5% after recovery from the acute process.	FVC% change HRCT fibrosis % change [Time frame: 24 weeks]	FVC stability or improvement Decreased oxygen requirement for physical activity Improved exercise capacity Visits to the Emergency or DH for respiratory causes Lung transplantation Death
NCT04461587 [49]	Pneumoconiosis	2	50	Interventional Open Label	Progressive fibrosis ** FEV1 ≤ 75%, or FVC ≤ 80%, or DLCO ≤ 70%, or abnormal 6MWT (oxygen desaturation ≥ 4% of resting at screening or within the past 6 months)	Change in FVC [Time frame: 12 months]	Changes in FEV1, DLCO, and 6MWT Changes in HRCT Inflammatory biomarkers SGRQ
NCT04193592 [50]	HPS-ILD	2	50	Interventional Open Label	Fibrotic abnormality affecting more than 5% of the lung parenchyma. Stable dose of corticosteroids. No cytotoxic, immunosuppressive, cytokine-modulating, or receptor antagonist agents were used in treatment.	Change in FVC [Time Frame: baseline, 6–12 months]	Changes in DLCO, FVC, and AE SAE
NCT03902509 [51]	Radiation-induced lung injury	2	126	Interventional Open Label	18–75 years old. Course of radiation-induced lung injury < 2 months. ECOG 0–2 Capable of eating solid food upon enrolment.	Change in DLCO% [Time frame: 8–24 weeks]	Changes in radiation-induced lung injury Score-change in HRCT Increase in effective lung volume Change in grade of cough, dyspnea, and fever
NCT03857854 [52]	Dm-ILD	3	152	Interventional Quadruple Blind	40% < FVC < 80% predicted. 30% < DLCO < 89% predicted Glucocorticoid and immunosuppressive therapy for more than 3 months	Change in FVC % [Time frame 52 weeks]	N/A
NCT03856853 [53]	SSc-ILD	3	144	Interventional Quadruple Blind	18–75 years old. SSc disease onset within 5 years. 40% < FVC 70% predicted.	Change in FVC % [Time frame 52 weeks]	N/A
NCT03385668 [54]	Pulmonary Fibrosis with MPO	2	7	Interventional Open Label	Possible UIP or NSIP based on HRCT. Pulmonary fibrosis refractory (according to the investigator’s judgment) to a conventional regimen used for anti-MPO associated vasculitis	Change in FVC % [Time frame 52 weeks]	AE Change in FVC% Change in DLCO% Change in 6MWT distance Progression-free survival Changes in dyspnea Changes in HRCT HAQ SF-36
NCT03315741 [55]	BOS	1	30	Interventional Open Label	Presence of cGVHD in an organ other than the lung Decrease in %FVC and/or %FEV1 ≥ 20% at screening compared with the pre-transplant baseline. Bronchodilator response on PFT testing those results in an FEV1 < 75% Life expectancy > 6 months	Number of participants that do not require a reduction in drug dose for more than 21 days due to AEs [Time frame 52 weeks]	AE The number of patients who experience treatment-emergent deaths during the study period and for 28 days after the last dose of study treatment. All-cause mortality BMI
NCT03221257 [56]	SSc-ILD	2	51	Interventional Triple blind	Grade ≥ 2 on the Magnitude of Task component of the MMDI FVC% ≤ 85% Onset of the first non-Raynaud manifestation of SSc within the prior 84 months. The presence of any GGO on HRCT	Change in FVC % [Baseline to 18 months, measured at 3-month intervals]	mRSS Changes in FVC and DLCO TDI SHAQ SGRQ HRCT measurements of quantitative lung fibrosis AEs
NCT02958917 [57]	PF-CHP	2	40	Interventional Double Blind	FVC ≥ 40%, DLCO ≥ 30%. Progressive fibrosis ***	Change in FVC % [Time frame 52 weeks]	Progression-free survival Change in DLCO% Proportion of patients with all-cause mortality, all-cause hospitalization, hospitalization for a respiratory cause, respiratory exacerbations requiring hospitalizations, evidence of progression in fibrosis on HRCT
NCT02496182 [58]	CHP	3	60	Interventional Quadruple Blind	CHP with a recent diagnosis confirmed by HRCT with or without biopsy	Change in FVC [Time frame 52 weeks]	6MWD San George Qty Score, SOBQ, and EQ5D Quality Scores Pulmonary artery systolic pressure with an echocardiogram Oxygen desaturation in exercise
NCT02262299 [59]	BOS	2–3	90	Interventional Triple Blind	Azithromycin therapy for ≥4 weeks prior to study start At least 6 months after transplantation BOS grades 1–3 Progressive disease ****	Change in FEV1 [Time frame 6 months]	Number of patients with treatment failure Change in BOS grade Change in pulmonary function parameters Change in 6MWTD All-cause hospital admission Death or re-transplantation rates Changes in EQ5D

Abbreviations: CTD-ILD: connective tissue disease, interstitial lung disease; RA: rheumatoid arthritis; IIM: idiopathic inflammatory myositis; SSc: systemic sclerosis; UCTD: undifferentiated connective tissue disease; IPAF: interstitial pneumonia with autoimmune features; ILD: interstitial lung disease; FVC: forced vital capacity; DLCO: diffusion capacity for carbon monoxide; Hb: hemoglobin; MMF: mycophenolate mofetil; TAC: tacrolimus; JAKi: Janus kinase inhibitor; CTX: cyclophosphamide; LEF: leflunomide; AZA: azathioprine; CTD: connective tissue disease; TLC: total lung capacity; DMARDs: disease-modifying antirheumatic drugs; CRP: C reactive protein; ERS: erythrocyte sedimentation rate; SAE: severe adverse event; HRCT: high-resolution computed tomography; PET-CT: positron emission tomography-computed tomography; FFDG: fluorine-18-2-fluoro-2-deoxy-D-glucose; AE: adverse event; AR: adverse reaction; SUSAR: serious and unexpected adverse reactions; EQ-5D: 5L EuroQol-5 Dimension 5 Levels; ARDS: acute respiratory distress syndrome; ICU: intensive care unit; SF-36: Short Form Health Survey 36; PF-CHP: progressive fibrosis chronic hypersensitivity pneumonitis (CHP): chronic hypersensitivity pneumonitis; 6MWD: six-minute walking distance; 6MWT: six-minute walk test. SGRQ: Saint George Respiratory Questionnaire; HPS: Hermansky Pudlak Syndrome; ECOG: Eastern Cooperative Oncology Group; ANC: absolute neutrophil count; PLT: platelets; TBIL: total bilirubin; BUN: blood urea nitrogen; ULN: upper limit of normal; ALT: Alanine Transaminase; AST: Aspartate Transferase; DM-ILD: Dermatomyositis Interstitial Lung Disease; MPO: Myeloperoxidase; UIP: usual interstitial pneumonia; NSIP: non-specific interstitial pneumonia; BOS: bronchiolitis obliterans syndrome; PFR: pulmonary function test; cGVHD: chronic graft-versus-host disease; SHAQ: Health Assessment Questionnaire Modified for Scleroderma; TDI: Mahler Modified Transitional Dyspnea Index; F-CHP: fibrotic chronic hypersensitivity pneumonitis; VAS: visual analogue scale; DH: day hospital; BMI: body mass index; MMDI: Mahler Modified Dyspnea Index; SDBQ: San Diego Shortness of Breath Questionnaire; EQ5D: EuroQol-5 Dimension. Notes: * >10% extent of fibrosis on HRCT scan; absolute decline in FVC% predicted >5% within the previous 6 months despite conventional treatment. ** Loss of lung function defined by decline in FEV1 or FVC or DLCO of 5% within the past 36 months prior to enrolment. *** Worsening respiratory symptoms and an increase in the extent of fibrosis on HRCT or a relative decline in the FVC% of at least 5%. Able to walk ≥100 m during the 6MWT at screening. **** total decline ≥ 200 mL in FEV1 and a decline ≥ 50 mL in the last two measurements.

## Data Availability

The data presented in this study are available on request from the corresponding author.

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
