# Peer review of "Efficacy of Pirfenidone and Nintedanib in Interstitial Lung Diseases Other than Idiopathic Pulmonary Fibrosis: A Systematic Review"

_ijms, 2023, doi:10.3390/ijms24097849_

Round 1

Reviewer 1 Report

My minor comments: 

In the SENSCIS trial, the rate of FVC decline was only about 40 mL/year (page 9, line 239) – this sentence needs to be clarified because it was treatment with nintedanib that slowed down the annual loss of FVC by 40 ml/year compared to placebo

Sometimes you write the names of the substances in capital "Nintedanib" or "Pirfenidone", other times in lower case: „nintedanib” or “pirfenidone”. It should be standardized - I think you can write it in lower case.

Also, the interchangeable use of the terms "progressive pulmonary fibrosing - PPF" and "progressive fibrosing ILD - PF-ILD" requires some explanation in the Introduction section.

The term "DMARDs treatment" needs to be expanded (line 382).

These comments are marginal. I find the whole work very valuable and well written.

Author Response

Reviewer 1:

General comment: I find the whole work very valuable and well written.

Response: We would like to thank the reviewer for her/his nice words.

Comment 1: In the SENSCIS trial, the rate of FVC decline was only about 40 mL/year (page 9, line 239) – this sentence needs to be clarified because it was treatment with nintedanib that slowed down the annual loss of FVC by 40 ml/year compared to placebo.

Response to comment 1: We thank the reviewer for her/his comment and we agree with her/him. Thus, we modify the sentence as follows “In the SENSCIS trial, treatment with nintedanib slowed down the annual loss of FVC by 40 ml/year compared to placebo, apparently unimpressive compared to the INBUILD trial [9,10].”

Comment 2: Sometimes you write the names of the substances in capital "Nintedanib" or "Pirfenidone", other times in lower case: „nintedanib” or “pirfenidone”. It should be standardized - I think you can write it in lower case.

Response to Comment 2: We thank the reviewer for her/his comment. We recognize that this comment is important. Accordingly, we modify the names within the text using lower case.

Comment 3: Also, the interchangeable use of the terms "progressive pulmonary fibrosing - PPF" and "progressive fibrosing ILD - PF-ILD" requires some explanation in the Introduction section.

Response to Comment 3: We thank the reviewer for her/his comment. We recognize that this comment is important. We standardize the use of PPF as the referral term within the text. Moreover, we add notes in the tables with explanation of the different definition of progressive fibrosis in the trials analyzed.

Comment 4: The term "DMARDs treatment" needs to be expanded (line 382).

Response to Comment 4: We thank the reviewer for her/his comment. We recognize that this comment is important. We explain the term DMARDs within the text and in the table.

Reviewer 2 Report

I appreciated the hard work and efforts of the authors in this study. 

I have some comments

1. Most references such as nintedanib included patients with PPF, so it is necessary to re-consider whether to set the target disease as non-IPF or PPF with the title or subject.

2.  Introduction [line 32] - sentence requiring expression correction: IPF is the most common cause?

3. Introduction [line 37] - Does the cited reference contain adequate information about pirfenidone to prevent acute exacerbation?  

4. Efficacy of Nintedanib [line 136] - It is necessary to mention more details about the ratio of enrolled patients using corticosteroids and immunosuppressants, and the difference in outcomes according to sub-group analysis.  

5. More details about other outcomes such as extrapulmonary symptoms need to be explained in addition to lung function. 

6. Unification of no-IPF or non-IPF is needed. 

7. The use of antifibrotic agents in patients with PPF or non-IPF is a hot and important topic nowadays. However, there are still many debates and challenges that have not yet been established, such as who needs it, when starts, and whether they use concomitant immunosuppressants. Please refer to the limitations of the antifibrotic use in this study. 

Author Response

Reviewer 2

General comment: I appreciated the hard work and efforts of the authors in this study. 

Response: We would like to thank the reviewer for her/his nice words.

Comment 1: Most references such as nintedanib included patients with PPF, so it is necessary to re-consider whether to set the target disease as non-IPF or PPF with the title or subject.

Response to comment 1: We thank the reviewer for her/his comment. We recognize that this comment is important. We included in the study ILD patients with a diagnostic label that is not IPF (Non-IPF). The PPF refers to a phenotype shared by several ILD patients with different underline diagnostic label. According to reviewer’s suggestion, we clarify this aspect in the text.

Comment 2: Introduction [line 32] - sentence requiring expression correction: IPF is the most common cause?

Response to comment 2: We thank the reviewer for her/his comment. We recognize that this comment is important. According to reviewer’s suggestion, we modify the sentence as follows: “Idiopathic pulmonary fibrosis (IPF) is the most common cause of idiopathic ILDs, and it is characterized by progressive fibrosis of the lungs with poor prognosis.”

Comment 3: Introduction [line 37] - Does the cited reference contain adequate information about pirfenidone to prevent acute exacerbation?  

Response to comment 3: We thank the reviewer for her/his comment. We recognize that this comment is important. We add a reference showing adequate information about effects on AE-IPF: “Petnak T, Lertjitbanjong P, Thongprayoon C, Moua T. Impact of Antifibrotic Therapy on Mortality and Acute Exacerbation in Idiopathic Pulmonary Fibrosis: A Systematic Review and Meta-Analysis. Chest. 2021;160(5):1751-1763. doi:10.1016/j.chest.2021.06.049”

Comment 4: Efficacy of Nintedanib [line 136] - It is necessary to mention more details about the ratio of enrolled patients using corticosteroids and immunosuppressants, and the difference in outcomes according to sub-group analysis.  

Response to comment 4: We thank the reviewer for her/his comment and we modify the paragraph according to reviewer’s suggestion as follows “In the INBUILD trial, glucocorticoids were taken by over half the patients at baseline, while 15.2% patient were taken immunomodulatory therapies [9]. The effect of nintedanib on reducing FVC decline was not influenced by the use of glucocorticoids and immunomodulatory therapies [21]. Several different types of ILD (other than IPF) were included in INBUILD and classified in five subgroups: hypersensitivity pneumonitis, idiopathic nonspecific interstitial pneumonia, unclassifiable ILD, autoimmune disease–related ILD, and “other” fibrosing ILDs. In a post-hoc analysis, no significant differences in efficacy between disease subgroups were observed [22].”

Comment 5: More details about other outcomes such as extrapulmonary symptoms need to be explained in addition to lung function. 

Response to comment 5: We thank the reviewer for her/his comment and we modify the discussion section according to reviewer’s suggestion as follows “Other respiratory and extra-respiratory parameters have been evaluated as secondary outcomes in the studies analyzed. Concerning lung function, there was a significant lower decline of diffusing lung capacity for carbon monoxide (DLCO) in the group receiving pirfenidone compared to placebo, suggestive of a beneficial treatment effect of pirfenidone [15,18,19]. However, it was not a uniform finding [16]. Of note, symptoms and quality of life measured by the Saint George Respiratory Questionnaire (SGRQ), were not significantly improved by antifibrotic treatment [9,10,17,19,20]. Similarly, nintedanib has not shown improvement in fibrosis skin involvement measured by the modified Rodnan skin score [10].”

Comment 6: Unification of no-IPF or non-IPF is needed. 

Response to comment 6: We thank the reviewer for her/his comment and we modify the text according to reviewer’s suggestion. We unify in non-IPF.

Comment 7: The use of antifibrotic agents in patients with PPF or non-IPF is a hot and important topic nowadays. However, there are still many debates and challenges that have not yet been established, such as who needs it, when starts, and whether they use concomitant immunosuppressants. Please refer to the limitations of the antifibrotic use in this study. 

Response to comment 7: We thank the reviewer for her/his comment. We recognize that this comment is important, and we modify the conclusion section as follows “However, results on both drugs should be interpreted with caution because of limitations in the available evidence. Moreover, there are several controversial points that should be clarified with further studies and evidence. Examples of some challenges that need to be addressed in the future are the timing of therapy initiation, and the strategies that should be adopted for overlap or combination with existing immunosuppressive therapies and potential drug interactions.”

Reviewer 3 Report

The topic of paper is of great interest. The paper is broad and detailed.

The paper is well structured and facilitates the reading and the follow-up of the study.

The systematic review carried out is correct, it is very well presented and structured. The study is perfectly reproducible, and the results obtained are of immediate clinical application.

Methodologically the study is well supported and explained.

The discussion is very well presented and contrasted with the bibliography commented on in the review carried out.

The conclusions are clear and fit the stated objectives.

I only have a few minimal considerations to make.

In my opinion, the use of acronyms in the title of paper and in the abstract are inappropriate since they can introduce some difficulties in the search for the topic that is consulted in the free access search and database. It is true that keywords make it easier to search for the subject, but in the title and in the abstract we must avoid using acronyms whenever possible.

Author Response

Reviewer 3

General comment: The topic of paper is of great interest. The paper is broad and detailed. The paper is well structured and facilitates the reading and the follow-up of the study. The systematic review carried out is correct, it is very well presented and structured. The study is perfectly reproducible, and the results obtained are of immediate clinical application. Methodologically the study is well supported and explained.  The discussion is very well presented and contrasted with the bibliography commented on in the review carried out.The conclusions are clear and fit the stated objectives.

Response: We would like to thank the reviewer for her/his nice words.

Comment 1: In my opinion, the use of acronyms in the title of paper and in the abstract are inappropriate since they can introduce some difficulties in the search for the topic that is consulted in the free access search and database. It is true that keywords make it easier to search for the subject, but in the title and in the abstract we must avoid using acronyms whenever possible.

Response to comment 1: We thank the reviewer for her/his comment. We recognize that this comment is important. Accordingly, we modify the title and abstract avoiding the use of acronyms.